# Radiative Energy Budget for East Asia Based on GK-2A/AMI Observation Data

Il-Sung Zo [1], Joon-Bum Jee [2,*], Kyu-Tae Lee [1], Kwon-Ho Lee [1,3], Mi-Young Lee [1] and Yong-Soon Kwon [1]

[1] Research Institute for Radiation-Satellite, Gangneung-Wonju National University (GWNU), Gangneung, 7, Jukheon-gil, Gangneung 25457, Gangwon, Republic of Korea

[2] Research Center for Atmosphere and Environment, Hankuk University of Foreign Studies (HUFS), 81, Oaedae-ro, Yonginm 17035, Gyeonggi, Republic of Korea

[3] Department of Atmospheric and Environmental Sciences, Gangneung-Wonju National University (GWNU), Gangneung, 7, Jukheon-gil, Gangneung 25457, Gangwon, Republic of Korea

* Correspondence: rokmcjjb717@hufs.ac.kr; Tel.: +82-31-8020-5586

**Abstract:** The incident and emitted radiative energy data for the top of the atmosphere (TOA) are essential in climate research. Since East Asia (11–61°N, 80–175°E) is complexly composed of land and ocean, real-time satellite data are used importantly for analyzing the detailed energy budget or climate characteristics of this region. Therefore, in this study, the radiative energy budget for East Asia, during the year 2021, was analyzed using GEO-KOMPSAT-2A/Advanced Metrological Imager (GK-2A/AMI) and the European Centre for Medium-range Weather Forecasts reanalysis (ERA5) data. The results showed that the net fluxes for the TOA and surface were $-4.09$ W·m$^{-2}$ and $-8.24$ W·m$^{-2}$, respectively. Thus, the net flux difference of 4.15 W·m$^{-2}$ between TOA and surface implied atmospheric warming. These results, produced by GK-2A/AMI, were well-matched with the ERA5 data. However, they varied with surface characteristics; the atmosphere over ocean areas warmed because of the large amounts of longwave radiation emitted from surfaces, while the atmosphere over the plain area was relatively balanced and the atmosphere over the mountain area was cooled because large amount of longwave radiation was emitted to space. Although the GK2A/AMI radiative products used for this study have not yet been sufficiently compared with surface observation data, and the period of data used was only one year, they were highly correlated with the CERES (Clouds and the Earth's Radiant Energy System of USA), HIMAWARI/AHI (Geostationary Satellite of Japan), and ERA5 data. Therefore, if more GK-2A/AMI data are accumulated and analyzed, it could be used for the analysis of radiant energy budget and climate research for East Asia, and it will be an opportunity to greatly increase the utilization of total meteorological products of 52 types, including radiative products.

**Keywords:** radiative energy budget; East Asia; Geo-KOMPSAT-2A; geostationary-Korean multi-purpose satellite-2A; GK-2A/AMI; ERA5; European center for medium-range weather forecasts reanalysis data

## 1. Introduction

Atmospheric and surface data are essential for analyzing the climate and weather of the earth's atmosphere, and the geostationary satellites data are used extensively because of the time-space continuity for these factors [1–5]. Notably, the global average temperature having increased by ~1 °C since 1850 (the preindustrial era) [6,7] has been pointed out as a cause of anthropogenic fossil fuels [8], so that the importance of observation data is being emphasized for accurate climate analyses [9–14].

Recently, IPCC's 6th Assessment Report (IPCC AR6) [15] analyzed the global energy budget using results from the 6th phase of the Climate Coupled Model Intercomparison Project (CMIP6) [16,17]. Although the overall accuracy has improved when compared to CMIP5 [18,19], these global results could not correctly be applied to specific regions due

to their relatively coarse resolutions (about 1° × 1°). In particular, East Asia (11–61°N, 80–175°E) is complexly composed of land and ocean (Figure 1), and the real-time satellite data observed in this region can be used for meteorological and climate research. Especially, the resolutions (2 km × 2 km) of radiative products produced by GK-2A/AMI which monitors East Asia, were better than CERES and ERA5 data (about 25 km × 25 km or 0.25° × 0.25°), and their accuracy was comparable to HIMAWARI/AHI which has similar characteristics to GK-2A/AMI. In addition, GK-2A/AMI products can be used diversely for accurate weather and climate analysis in East Asia, because GK-2A/AMI produced a total of 52 meteorological and climatic components, including radiative products. Therefore, in this study, the radiative energy budget for this region was analyzed using products of GK-2A/AMI, with high-performance meteorological sensors (http://nmsc.kma.go.kr/enhome/html/main/main.do (accessed on 1 February 2023)), as well as sensible and latent heat flux data (SHF and LHF), from ERA5 [20,21].

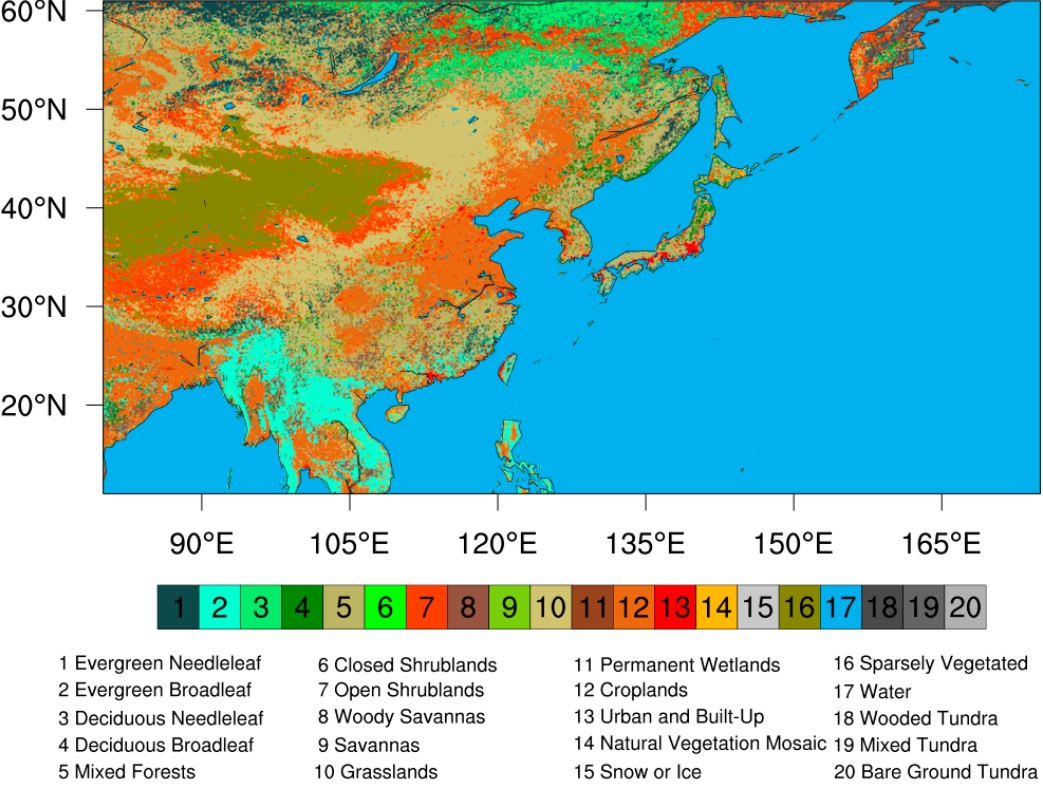

**Figure 1.** Landcover information within the research area (Data source: MODIS-IGBP 21 class LULC data [22]).

The GK-2A/AMI data used in this study, was compared with CERES and HIMAWARI satellite data to verify the accuracy. In addition, the analysis results of radiative energy budget for East Asia were compared with ERA5 data. Particularly, the radiative energy budgets for three regions (mountain, plain, and ocean) of different characteristics in East Asia were analyzed, and the results were compared with each other.

The remainder of this paper was written in the following structure: Section 2 introduced the data (GK2A/AMI and ERA5) and methods used in this study; Section 3 discussed the results analyzed by radiative energy budget for East Asia; and Section 4 provided the conclusion of the study.

## 2. Materials and Methods

### 2.1. Geo-KOMPSAT-2A (GK-2A) Data

The National Meteorological Satellite Center of the Korea Meteorological Administration (KMA) launched the GEO-KOMPSAT-2A/Advanced Meteorological Imager (GK-

2A/AMI) on 5 December 2018, and provided observation data shortly thereafter. However, the stability of radiative products were only established in September 2020. Accordingly, the data from 1 January to 31 December 2021 were analyzed in the present study. The observation data of GK-2A/AMI were divided into full-disk (FD), extended local area (ELA), and local area (LA), of which FD data were produced every 10 min, while ELA and LA data were produced every 2 min [23]. In this study, the ELA data were used for radiative energy budget analysis for East Asia because of its high spatiotemporal resolution.

　　GK-2A/AMI collects images of visible (shortwave) and infrared (longwave) wavelength regions, using 16 channels. The spatial resolution for each channel falls within the range of 0.5–2.0 km (Table 1), and produces 23 primary and 29 secondary meteorological products [24]. Amongst these GK-2A/AMI products, we used the following six types in this study: reflected shortwave radiation at TOA (RSR), downward shortwave radiation at surface (DSR), absorbed shortwave radiation at surface (SSR), outgoing longwave radiation at TOA (OLR), downward longwave radiation at surface (DLR), and upward longwave radiation at surface (ULR) [25].

**Table 1.** Specifications for GEO-KOMPSAT-2A Advanced Meteorological Imager (GK-2A/AMI) spectral channels.

| Channel Number | Channel Name | Wavelength (μm) | Resolution (km) |
| --- | --- | --- | --- |
| 1 | VIS (VIS0.4) | 0.4310–0.4790 | 1.0 |
| 2 | VIS (VIS0.5) | 0.5028–0.5175 | 1.0 |
| 3 | VIS (VIS0.6) | 0.6250–0.6600 | 0.5 |
| 4 | VNIR (VIS0.8) | 0.8475–0.8705 | 1.0 |
| 5 | SWIR (NR1.3) | 1.3730–1.3830 | 2.0 |
| 6 | SWIR (NR1.6) | 1.6010–1.6190 | 2.0 |
| 7 | MWIR (IR3.8) | 3.7400–3.9600 | 2.0 |
| 8 | MWIR (IR6.3) | 6.0610–6.4250 | 2.0 |
| 9 | MWIR (IR6.9) | 6.8900–7.0100 | 2.0 |
| 10 | MWIR (IR7.3) | 7.2580–74330 | 2.0 |
| 11 | TIR (IR8.7) | 8.4400–8.7600 | 2.0 |
| 12 | TIR (IR9.6) | 9.5430–9.717 | 2.0 |
| 13 | TIR (IR10.5) | 10.2500–10.6100 | 2.0 |
| 14 | TIR (IR11.2) | 11.0800–11.3200 | 2.0 |
| 15 | TIR (IR12.3) | 12.1500–12.4500 | 2.0 |
| 16 | TIR (IR13.3) | 12.2100–13.3900 | 2.0 |

　　The calculating methods of GK-2A/AMI products are as shown in Equations (1)–(6) below, and are summarized as follows: the RSR was calculated using the solar constant ($S_0$ = 1361 W·m$^{-2}$) [26], eccentricity ($E_0$), solar zenith angle ($SZA$), and TOA albedo ($\alpha$; Equation (1)) [27]. The TOA albedo was calculated using a regression equation, according to the narrowband reflectance for channels 1–6 of GK-2A/AMI, $SZA$, viewing zenith angle ($VZA$), relative azimuth angle ($RAA$), surface type, and the presence/absence of clouds [28], and the coefficients of the regression equation were calculated using the SBDART radiative transfer model [29]. The DSR was computed based on the presence/absence of clouds, and atmospheric transmittance ($\tau$) over land and ocean (Equation (2)) [30]; whereas the SSR was calculated based on DSR and surface albedo ($SAL$; Equation (3)) [31]. Here, as the SAL was computed from land only, the oceanic value was calculated using the Fresnel equations [32]:

$$RSR = S_0 \cdot E_0 \cdot cos(SZA) \cdot \alpha \tag{1}$$

$$DSR = S_0 \cdot E_0 \cdot cos(SZA) \cdot \tau \tag{2}$$

$$SSR = DSR \cdot (1 - SAL) \tag{3}$$

　　The OLR was calculated by converting the radiance, for each channel, into flux (irradiance) using channels 8, 12, 15, and 16 [33]. In Equation (4), *F* represents flux, subscripts indicate the channel number of GK-2A/AMI, and $c0$–$c8$ are the pre-calculated constants [34].

The DLR was calculated using the all sky emissivity ($\varepsilon_{allsky}$), and Stefan–Boltzmann's law (Equation (5)). Here, the $\varepsilon_{allsky}$ was calculated using a numerical weather prediction (NWP) model for the KMA, and cloud amount data from GK-2A [35]. The symbol of $T_{2m}$ in Equation (5) means the temperature at a height of 2 m. Lastly, the ULR was calculated using broadband emissivity at the surface ($\varepsilon_{LW}$), Stefan–Boltzmann's law, LST (land surface temperature, GK-2A/AMI product), SST (sea surface temperature, GK-2A/AMI product), and DLR (Equation (6)) [36–38]:

$$OLR = c0 + c1F_8 + c2F_8{}^2 + c3F_{12} + c4F_{12}{}^2 + c5ln(F_{15}) + c6ln(F_{15})^2 + c7F_{16} + c8F_{16}{}^2 \qquad (4)$$

$$DLR = (\varepsilon_{allsky})\sigma T_{2m}{}^4 \qquad (5)$$

$$ULR = \varepsilon_{LW}\sigma T^4 + (1 - \varepsilon_{LW})DLR \qquad (6)$$

For the verification of these GK-2A/AMI radiative products (RSR, DSR, SSR, OLR, DLR, and ULR), they were compared to Clouds and the Earth's Radiant Energy System of USA (CERES) satellite data from 1–30 September 2020, and the results are presented in Figure 2. The CERES data used for verification of the GK-2A/AMI radiative products were the Terra CERES single scanner footprint (SSF) Level 2 Edition 3A (Ed4A). The radiative products of GK-2A/AMI were produced every 2 min for the extended local area (ELA). However, since CERES is a polar orbital satellite, Figure 2 shows CERES pixels (20 km × 20 km), and the average of GK-2A/AMI pixels (2 km × 2 km) within ±10 min, for passing time of CERES. And since GK-2A/AMI radiative data observed every 10 min was used in this study, CERES data shown in Figure 2 were collected within ±5 min for GK-2A/AMI observation time. That is, for example, GK-2A/AMI products at 0110 UTC was compared to CERES 0105–0115 UTC data, and this method can perform a more detailed analysis than the previous methods [39–41], which assumed identical atmospheric condition every 30 min.

In Figure 2, DSR and SSR were related to atmospheric transmittance ($\tau$) according to clouds, as shown in Equations (2) and (3), and the parts with a large difference between the CERES and GK-2A/AMI were indicated. In other words, partly cloudy/aerosol was detected in GK-2A/AMI, but if not in CERES, the DSR and SSR values produced by GK-2A/AMI were smaller than CERES, and are corrected. Nevertheless, as shown in Figure 2, the correlation coefficients between the radiative products of GK-2A/AMI and CERES were more than 0.9, and the root mean square error (RMSE) was determined to be good.

Figure 2 shows the comparison of real-time observation data of GK-2A/AMI and CERES, and the purpose of this paper is to use GK-2A/AMI products for analyzing the weather and climate as well as radiative energy budget. Therefore, GK-2A/AMI data averaged at intervals of 1° × 1° were compared with HIMAWARI/AHI, CERES EBAF (Energy Balance and Filled) (https://ceres.larc.nasa.gov/data/ (accessed on 1 February 2023)), and ERA5, as shown in Table 2. The data periods of CERES EBAF and ERA5 used to calculate the values in Table 2 were January and July 2021, but in the case of HIMAWARI/AHI, the data during this period were not obtained, so the data in September 2020 were used.

GK-2A/AMI data should be verified in comparison with surface observation data. However, there is no BSRN (Baseline Surface Radiation Network, [42,43]) observation station belonging to the World Radiation Center (WRC) in Korea; thus, we could not get reliable real-time surface observation data. On the other hand, Japan had about five BSRN stations, and similar HIMAWARI/AHI products to GK-2A/AMI radiative products were calibrated with their surface observation data. Therefore, in this study, the radiative products of GK-2A/AMI and HIMAWARI/AHI were compared. As the results, correlation coefficients (R) between GK-2A/AMI and HIMAWARI/AHI for all of radiative products were equal to 0.99, and MBE and MPE were less than 3 W·m$^{-2}$ and 3%, respectively, as shown in Table 2.

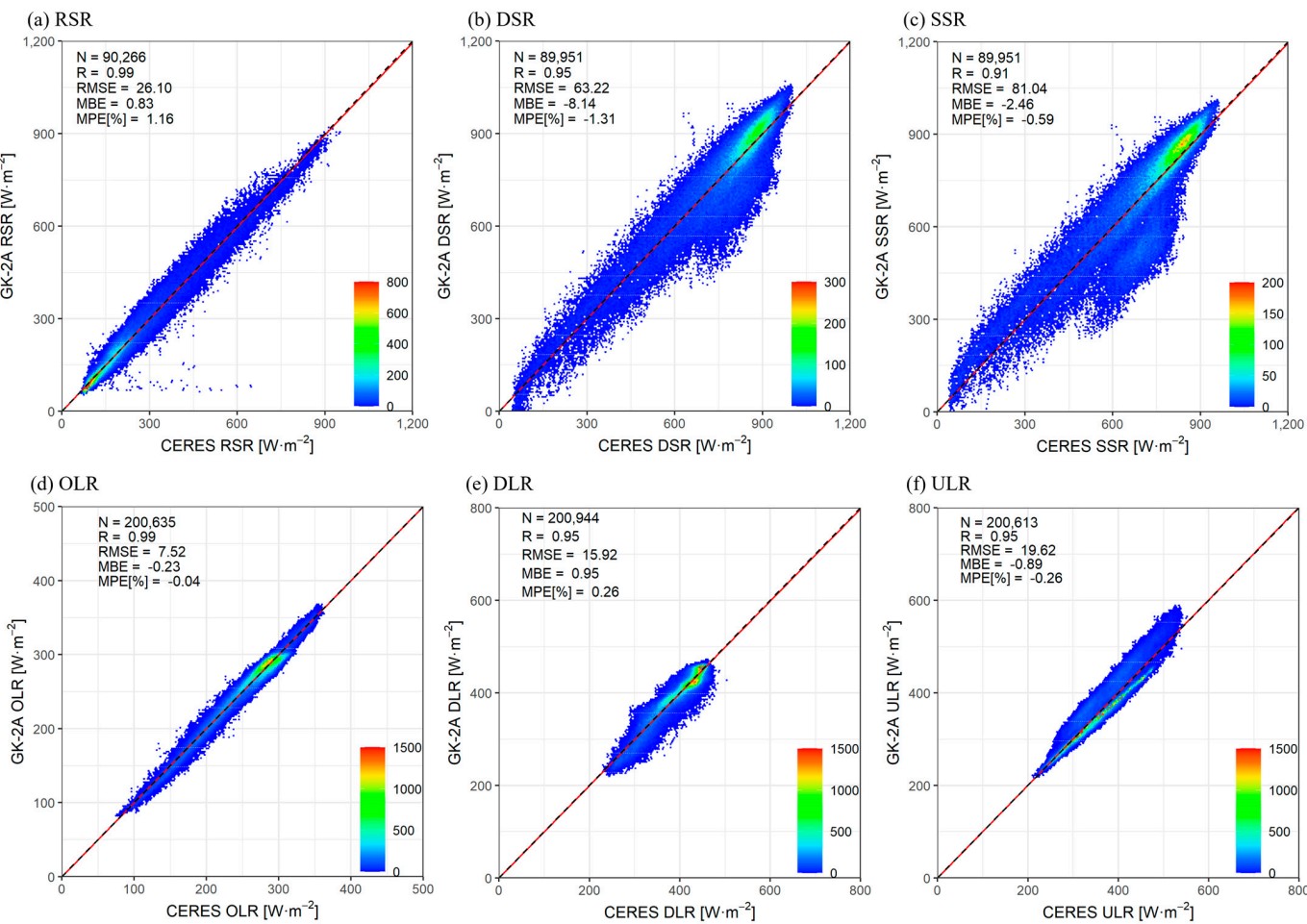

**Figure 2.** Accuracy analysis results for GK-2A/AMI radiative products: (**a**) RSR, (**b**) DSR, (**c**) SSR, (**d**) OLR, (**e**) DLR, and (**f**) ULR, using CERES satellite data. Data period from 1–30 September 2020.

**Table 2.** Statistical comparison values (R, MBE, and MPE) of HIMAWARI/AHI, CERES EBAF, and ERA5 data for GK-2A/AMI radiative products. Data period from 1–31 January and 1–31 July 2021, except but HIMAWARI/AHI from 1–30 September 2020.

| Radiative Products | Comparative Data | R | MBE (W·m$^{-2}$) | MPE (%) |
|---|---|---|---|---|
| RSR | HIMAWARI/AHI | 0.99 | −1.36 | −2.43 |
| | CERES EBAF | 0.97 | −3.74 | −5.04 |
| | ERA5 | 0.96 | −3.57 | −5.25 |
| DSR | HIMAWARI/AHI | 0.99 | −1.99 | 2.01 |
| | CERES EBAF | 0.98 | −2.17 | 2.75 |
| | ERA5 | 0.98 | −2.66 | −2.09 |
| SSR | HIMAWARI/AHI | 0.99 | −0.31 | 0.58 |
| | CERES EBAF | 0.98 | −0.49 | 0.71 |
| | ERA5 | 0.98 | −0.53 | 0.70 |
| OLR | HIMAWARI/AHI | 0.99 | 0.23 | 0.15 |
| | CERES EBAF | 0.98 | −0.46 | −0.19 |
| | ERA5 | 0.99 | −0.68 | −0.24 |
| DLR | HIMAWARI/AHI | 0.99 | 2.58 | 0.79 |
| | CERES EBAF | 0.99 | 2.97 | 0.90 |
| | ERA5 | 0.99 | 3.26 | 0.95 |
| ULR | HIMAWARI/AHI | 0.99 | 0.93 | 0.22 |
| | CERES EBAF | 0.99 | 1.52 | 0.44 |
| | ERA5 | 0.99 | 1.74 | 0.48 |

The values of R between the radiative products of GK-2A/AMI and CERES EBAF were more than 0.97, and those for the mean bias error (MBE) and mean percent error (MPE) were less than 4.0 W·m$^{-2}$ and 5.1%, respectively. Moreover, since the ERA5 radiative data were verified by the observation and model calculated values, these data were compared with GK-2A/AMI radiative products. As a result, all of the statistical values (R, MBE, and MPE) produced between ERA5 and GK-2A/AMI radiative products were determined to be excellent.

### 2.2. ERA5 Sensible and Latent Heat Data

In addition to radiative energy, the sensible and latent heat to be released or absorbed by surface are important factors for analysis of atmospheric energy balance. However, since these sensible and latent heat are not direct observation factors, they are not included in GK−2A/AMI satellite data. But ERA5 radiative data maintains the global energy balance with sensible heat flux (SHF) and latent heat flux (LHF) data [44,45], as shown in Table 3; it showed a high correlation with GK-2A/AMI products. Therefore, in this study, ERA5′s SHF and LHF data as well as GK-2A/AMI products were used for the analysis of regional energy budget for East Asia. The spatial resolution for ERA5 data was 0.25° × 0.25° with an hourly temporal resolution [46,47].

**Table 3.** Seasonal mean values for radiative components by GK2A/AMI (the values in parentheses are by CERES EBAF) and non-radiative components by ERA5 in research area, during 2021.

| Season | Radiative Component | | | | | | | Non-Radiative Component (ERA5) | |
| | Shortwave | | | | Longwave | | | | |
| | ISR | RSR | DSR | SSR | OLR | ULR | DLR | SHF | LHF |
|---|---|---|---|---|---|---|---|---|---|
| Spring | 395.66 (395.68) | 125.39 (128.99) | 218.32 (220.21) | 179.16 (179.88) | 234.70 (235.17) | 377.91 (376.12) | 315.51 (313.69) | 24.13 | 63.87 |
| Sumer | 452.36 (452.37) | 140.05 (144.18) | 225.72 (227.69) | 200.39 (200.91) | 238.83 (239.31) | 426.46 (424.83) | 373.13 (370.94) | 21.10 | 78.58 |
| Autumn | 280.83 (280.85) | 89.87 (91.16) | 141.85 (142.74) | 123.95 (124.29) | 232.57 (233.04) | 386.94 (385.31) | 328.83 (326.84) | 15.77 | 81.54 |
| Winter | 213.34 (213.36) | 70.77 (72.31) | 114.23 (115.33) | 95.21 (95.67) | 226.37 (226.81) | 337.42 (336.09) | 276.19 (274.68) | 23.79 | 87.83 |
| Mean | 335.55 (335.56) | 106.52 (109.16) | 175.03 (176.49) | 149.68 (150.18) | 233.12 (233.58) | 382.18 (380.58) | 323.42 (321.53) | 21.20 | 77.96 |

### 2.3. Equations for Radiative Energy Budget

The symbols (NSF_TOA, NLF_TOA, NSF_Sfc, and NLF_Sfc) used for TOA and surface in this paper are defined in the following Equations (7)–(10).

$$NSF\_TOA = ISR - RSR \tag{7}$$

$$NLF\_TOA = -OLR \tag{8}$$

$$NSF\_Sfc = SSR \tag{9}$$

$$NLF\_Sfc = DLR - ULR \tag{10}$$

Net flux (NF) in this study was calculated as the difference between the incoming and outgoing flux; whereas NF_TOA refers to the TOA net flux, and was calculated as the difference between incoming solar radiation (ISR; Equations (11) and (12)) and outgoing radiation (RSR and OLR). The "+/−" signs for the values of radiative and non-radiative components used in this paper indicate downward and upward direction of radiation, respectively. Additionally, the surface net flux (NF_Sfc) and atmosphere net flux (NF_Atm) were calculated on the basis of non-radiative components such as SHF and LHF, as well as SSR, DLR, ULR, and NF_TOA (Equations (13) and (14)) [48–50]:

$$NF\_TOA = ISR \text{ - } RSR - OLR \tag{11}$$

$$ISR = S_0 \cdot E_0 \cdot cos(SZA) \tag{12}$$

$$NF\_Sfc = SSR + DLR - ULR - SHF - LHF \tag{13}$$

$$NF\_Atm = NF\_TOA - NF\_Sfc \tag{14}$$

## 3. Results and Discussion

### 3.1. Monthly Variation of Radiative Energy Budget Components for East Asia

Hourly data from GK-2A/AMI radiative products, for the period of 1 January to 31 December 2021, were used to analyze the radiant energy budget for East Asia (11–61°N, 80–175°E). The seasonal means for six radiative products, as well as the SHF and LHF for this period are shown in Table 4, and the values of CERES EBAF in parentheses are displayed together to compare the values of GK-2A/AMI products. In shortwave components, the incoming solar radiation (ISR) at the TOA reached a maximum of 452.36 W·m$^{-2}$ in summer, and minimum of 213.34 W·m$^{-2}$ in winter due to the variations in solar zenith angles. Furthermore, the RSR, which reflected off the surface and atmosphere into outer space, showed similar seasonal variations as ISR. Therefore, the net solar radiation at the TOA (NSF_TOA) was expressed as the difference between ISR and RSR, with an annual mean value of 229.03 W·m$^{-2}$. Consequently, the annual mean albedo for incoming solar radiation is equal to 0.32.

DSR is the solar radiation that transmitted toward the surface from the atmosphere, and its seasonal variation was similar to that of ISR and RSR, with an annual mean value of 175.03 W·m$^{-2}$. Most of DSR was absorbed by the surface (SSR; 149.68 W·m$^{-2}$), and consequently net solar radiation (NSF_Atm; 229.03 − 149.68 = 79.35 W·m$^{-2}$) accumulated in the atmosphere was used to heat the atmosphere.

As downward longwave radiation at TOA was considered to be zero, net longwave radiation at TOA (NLF_TOA) was expressed as OLR, which is emitted from the surface and atmosphere into the outer space. As shown in Table 4, the monthly variation of OLR was not significant. The net flux at the TOA (NF_TOA; −4.09 W·m$^{-2}$)was calculated as the difference between the net TOA solar radiation (NSF_TOA; 229.03 W·m$^{-2}$) coming into the earth's atmosphere and net TOA longwave radiation (NLF_TOA; 233.12 W·m$^{-2}$) emitted back to the space, and this value is used as an important factor in determining the climate of a region.

Upward longwave radiation (ULR) emitted from the surface varied depending on surface characteristics and temperature, with maximum and minimum values of 426.46 W·m$^{-2}$ and 337.42 W·m$^{-2}$, in summer and winter, respectively. Notably, the seasonal variation of ULR was influenced by the specific heat of surfaces, depending on the type of surface, and the DLR emitted from the atmosphere to the surface was substantially influenced by the atmospheric temperature and cloud cover, similar to ULR, producing maximum (370.94 W·m$^{-2}$) and minimum (276.19 W·m$^{-2}$) values in summer and winter, respectively. Therefore, the net longwave radiation (NLF_Sfc) entering from the surface to the atmosphere is the difference between the ULR and DLR, which was −58.76 W·m$^{-2}$, and this value is important for the evaluation of greenhouse effect because it will be absorbed by gases and clouds in the atmosphere and re-emitted to the surface.

In Table 5, SHF and LHF data of ERA5 were included to analyze the dynamic effects between the surface and atmosphere, since they were important factors for total atmospheric energy balance as well as radiative energy. Both SHF and LHF are significantly influenced by space and time, and exhibited different characteristics from monthly changes of radiative products, as shown in Table 3. Specifically, the SHF was largely influenced by differences in temperature between the surface and atmosphere, wind speed, and surface roughness [51,52]; whereas the LHF was calculated based on evaporation and wind speed, near the surface [53]. Figure 3 shows the monthly mean values for SHF and LHF over East Asia, during January and July 2021. The SHF recorded maximum values of more

than 200 W·m$^{-2}$ (Figure 3a) in January, near the sea around the Korean Peninsula, and Japan (which differed significantly from the values on land). Alternatively, during July, a maximum value for SHF of 90 W·m$^{-2}$ occurred over mainland China, and the SHF over the open ocean was less than 30 W·m$^{-2}$ (Figure 3b). LHF maximum values of more than 200 W·m$^{-2}$ were recorded over the open ocean near Japan, and near the equator, in January and July, respectively. As a result, maximum values for monthly SHF and LHF, for East Asia, were shown over the open ocean during winter (Figure 3a,c), where the annual means were 21.2 W·m$^{-2}$ and 77.96 W·m$^{-2}$ (Table 5), respectively. The ratios of the annual mean SHF and LHF for ISR were 6.32% and 23.23%, respectively, which were slightly different from the global means (10% and 20%, respectively) [54,55].

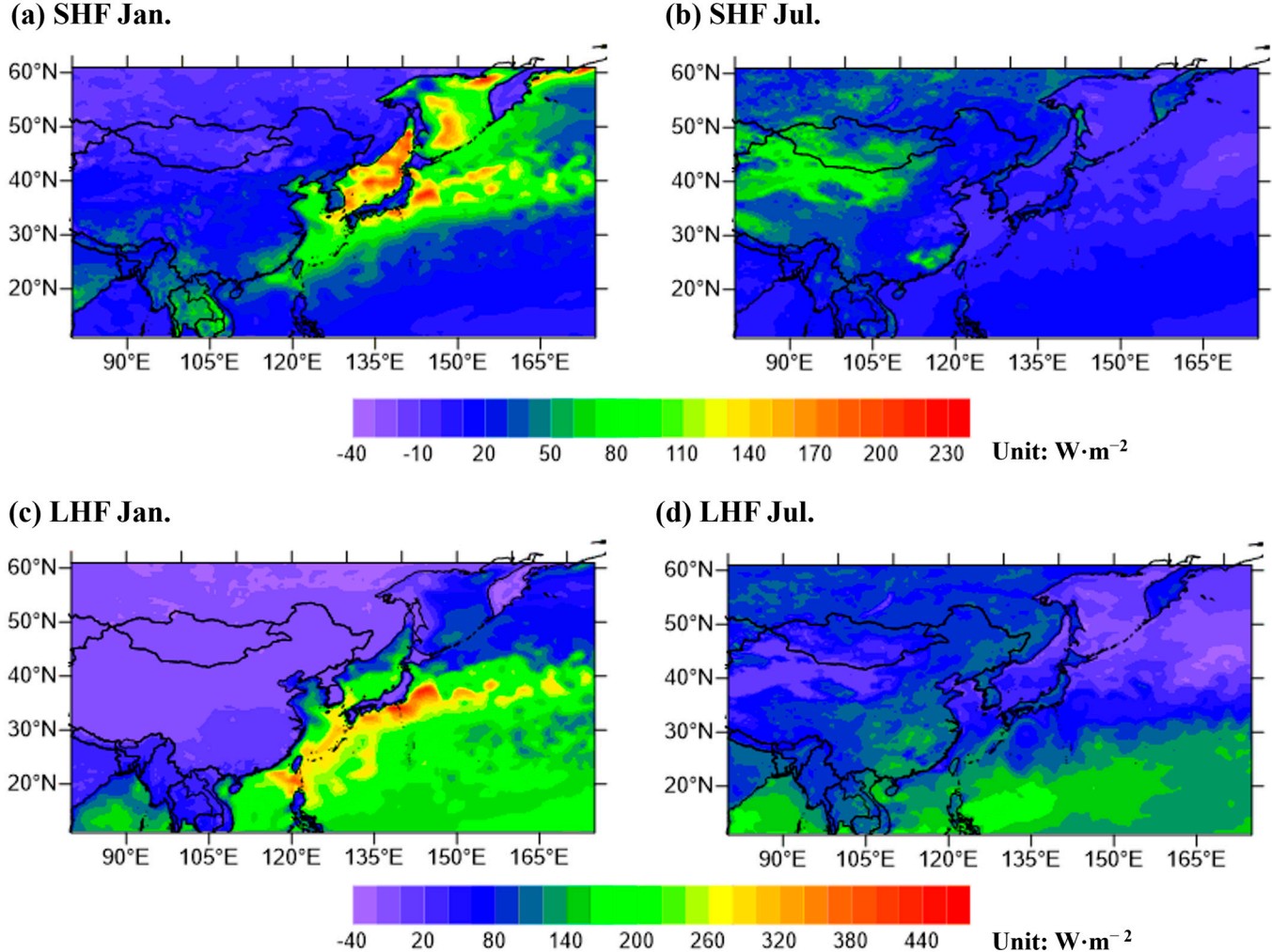

**Figure 3.** Monthly means for sensible and latent heat flux (SHF, LHF) during January and July 2021, from ERA5.

**Table 4.** Comparisons of radiative energy budgets calculated by GK-2A/AMI and ERA5. Units for Radiative Energy Budget Components: W·m$^{-2}$.

| Radiative Energy Budget Component | GK-2A/AMI | ERA5 Data |
|---|---|---|
| NF_TOA | −4.09 | −4.01 |
| NF_Atm | −4.15 | −4.05 |
| NLF_Sfc | −8.24 | −8.06 |

The monthly variations in net fluxes at the TOA and surface (NF_TOA and NF_Sfc, respectively) are illustrated in Figure 4. The NF_TOA represented a positive value from April to September, and a negative value from October to the following March, with variations in the solar zenith angle. The NF_Sfc was similar to NF_TOA; however, the maximum value for NF_TOA occurred in June (91.66 W·m$^{-2}$), while the same for NF_Sfc was recorded in May (58.60 W·m$^{-2}$), according to the surface and atmospheric properties in East Asia. Consequently, the net flux for the atmosphere (NF_Atm) over East Asia, which was represented by the difference between NF_TOA and NF_Sfc, had a positive value from April to September, and a negative value from October to the following March. The value of NF_Atm was the highest in June (33.41 W·m$^{-2}$), while the lowest (−21.76 W·m$^{-2}$) was observed in November since SHF values were low and relatively consistent due to the clear and dry weather over East Asia at that time.

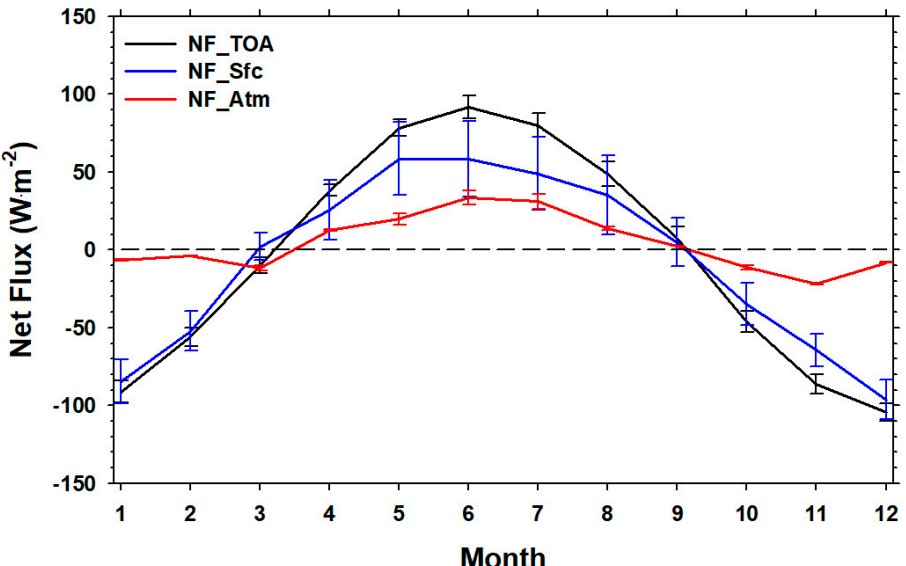

**Figure 4.** Variation of monthly mean net flux during 2021 (black, blue, and red lines represent the top of atmosphere, surface, and atmosphere, respectively. The vertical bars in this figure represent the standard deviation, but their values are too small. So, they are drawn by magnifying them five times).

### 3.2. Comparisons of Radiative Energy Budget Components Calculated by GK-2A/AMI and ERA5 for East Asia

This section was written for verifying the accuracy of GK-2A/AMI products. Therefore, the radiative energy budget components calculated by GK-2A/AMI and ERA5 were compared, and the results are shown in Table 4. As shown at the bottom of Table 2, annual mean of net longwave radiation values (NLF_Sfc) for the surface of this study area were calculated by adding the SHF (−21.20 W·m$^{-2}$) and LHF (−77.96 W·m$^{-2}$) to the differences (−58.76 W·m$^{-2}$) between ULR (382.18 W·m$^{-2}$) and DLR (323.42 W·m$^{-2}$). This resulted in a value of −157.92 W·m$^{-2}$, while the NF_Sfc (−8.24 W·m$^{-2}$) was the sum of NLF_Sfc (−157.92 W·m$^{-2}$) and NSF_Sfc (149.68 W·m$^{-2}$). Consequently, the difference of −4.15 W·m$^{-2}$ between the net flux at the surface (NF_Sfc, −8.24 W·m$^{-2}$) and that at the TOA (NF_TOA; 335.55 − 106.52 − 233.12 = −4.09 W·m$^{-2}$) was attributed to the heat in the atmosphere of East Asia.

The net radiation at the top of the atmosphere depends on latitude, which has a positive value at the equator, but changes to a negative value toward the pole [56]. Therefore, during the period of this study, the global radiative energy budget (90°S–90°N, 0°W–180°E) by ERA5 showed a value of 0.89 W·m$^{-2}$ at the top of the atmosphere, which was almost balanced, but the East Asia region (11–61°N, 80–175°E) corresponding to this study had a value of −4.01 W·m$^{-2}$, as shown in Table 4, and the difference from this study (−4.09 W·m$^{-2}$) by GK-2A/AMI was only 0.08 W·m$^{-2}$. Additionally, the difference in the radiative energy budget (NF_Sfc), including the SHF and LHF, between ERA5 and this

study was 0.18 W·m$^{-2}$. Therefore, the radiative products by GK-2A/AMI were evaluated to match well with the ERA5 reanalysis data.

However, this result was only based on data collection during one year (2021), and thus may not be appropriate for the detailed analysis of radiative energy budget and climate in East Asia. Therefore, if GK-2A/AMI products are accumulated for several years, it could be used for more detailed radiation and climate studies for East Asia region, including the Korean Peninsula.

### 3.3. Regional Distribution of Radiative Energy Budget Components for East Asia

The annual mean values (2 km × 2 km intervals) for the regional characteristics analysis of GK-2A/AMI radiative products (10 min intervals) are presented in Figure 5. Figure 5a shows the values for ISR at TOA, which varied from low latitude to middle latitude by about 420~230 W·m$^{-2}$ in this research area. Figure 5b shows the RSR, which represented the solar radiation reflected by the surface, atmosphere, and clouds. In Figure 5b, a maximum value for RSR of more than 130 W·m$^{-2}$ appeared in the area affected by the leeward clouds of the Himalayas, whereas minimum values of less than 70 W·m$^{-2}$ were recorded over the equatorial pacific region. Furthermore, the values for DSR reaching the surface are typically about 50% of the global ISR values, and this trend was consistent with the values shown in Figure 5c. Specifically, DSR values of more than 200 W·m$^{-2}$ are shown at low latitude or high altitudes region such as the Himalayas, and the values of less than 100 W·m$^{-2}$ are shown at high latitudes. The SSR values in Figure 5d are related to surface albedo, but had similar spatial distributions to DSR.

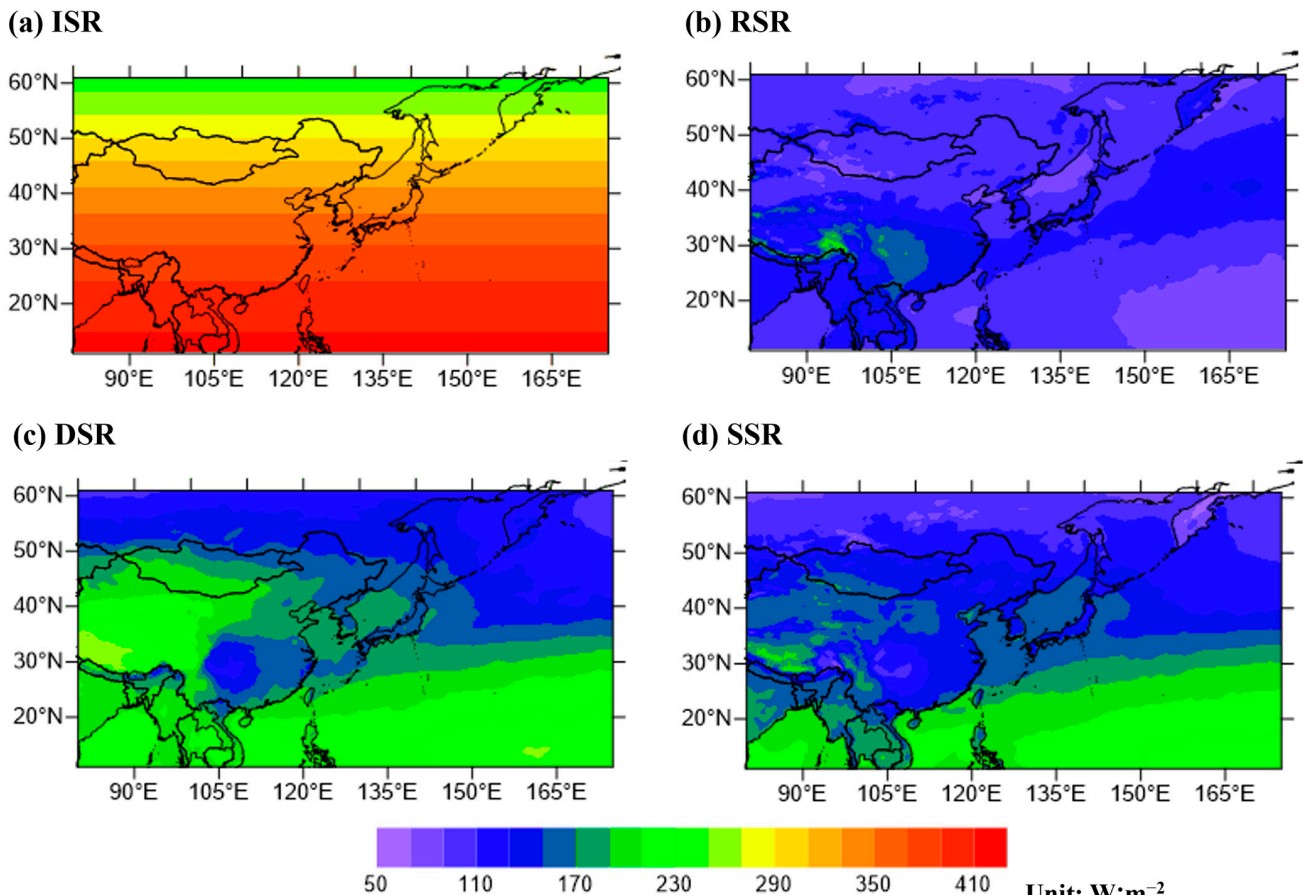

**Figure 5.** *Cont.*

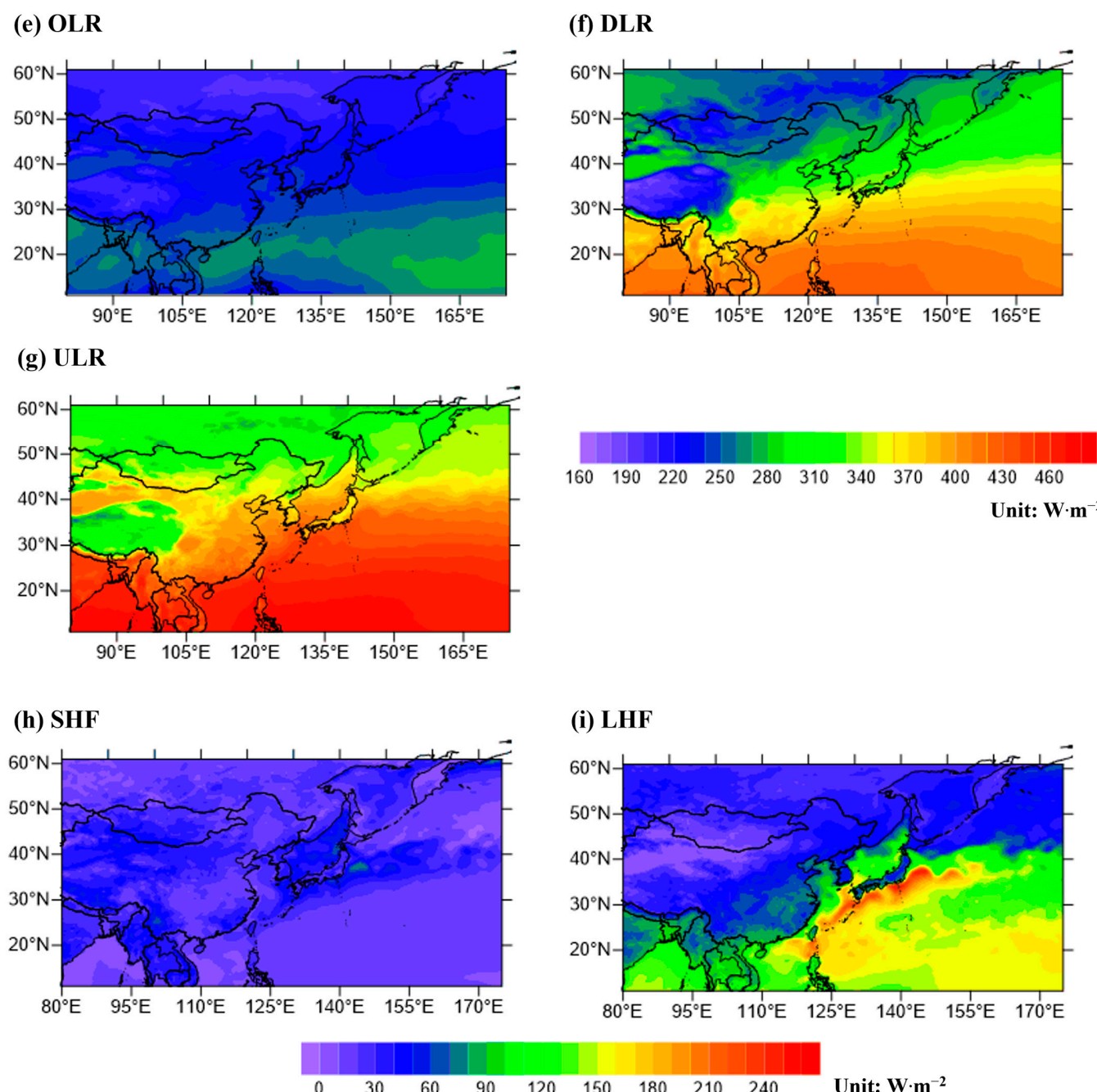

**Figure 5.** Annual mean values of (**a**) ISR, (**b**) RSR, (**c**) DSR, (**d**) SSR, (**e**) OLR, (**f**) DLR, (**g**) ULR, (**h**) SHF and (**i**) LHF over East Asia region for the year of 2021.

The OLR emitted to space from TOA varied depending on the surface and atmospheric temperature, and clouds. Additionally, maximum values of more than 250 W·m$^{-2}$ were shown over the equatorial Pacific region (Figure 5e). The DLR emitted from the atmosphere to the surface varied with the temperature of atmosphere and clouds, and the ULR was also released into the atmosphere according to the temperature of the surface. They exhibited similar spatial distributions as shown in Figure 5f,g.

The temperature and evaporation rates of surface are important factors in the calculation of SHF and LHF. The annual mean value for SHF reached a maximum of more than 70 W·m$^{-2}$ over the ocean near the Korean Peninsula and Japan (Figure 5h), and similar values were exhibited on land near southwest China as well as the leeward side of the Himalayas. As evaporation rate is an important factor in variation of LHF, the values of

LHF were more distinct over the ocean compared to land (Figure 5i). And maximum LHF values of more than 200 W·m$^{-2}$ were recorded above the southern ocean of Japan, where they were twice as large as those recorded for the SHF.

*3.4. Geographical Characteristics of Radiative Energy Budget Components*

East Asia is characterized by various types of landcover (Figure 1). Therefore, the energy budget components were analyzed for three regions (0.25° × 0.25° for each of A, B and C), as shown in Figure 6. The central latitude for these regions were equal to 37.75°N, and the longitudes of mountain (A), plain (B), and ocean (C) were 88.25°E, 127.00°E, and 157.00°E, respectively. The area A was located in the Tibetan Plateau, which was more than 5000 m above sea level, where temperatures were low. Area B lay in the plains region of South Korea (~50 m above sea level), and area C lay in the Northern Pacific (at sea level).

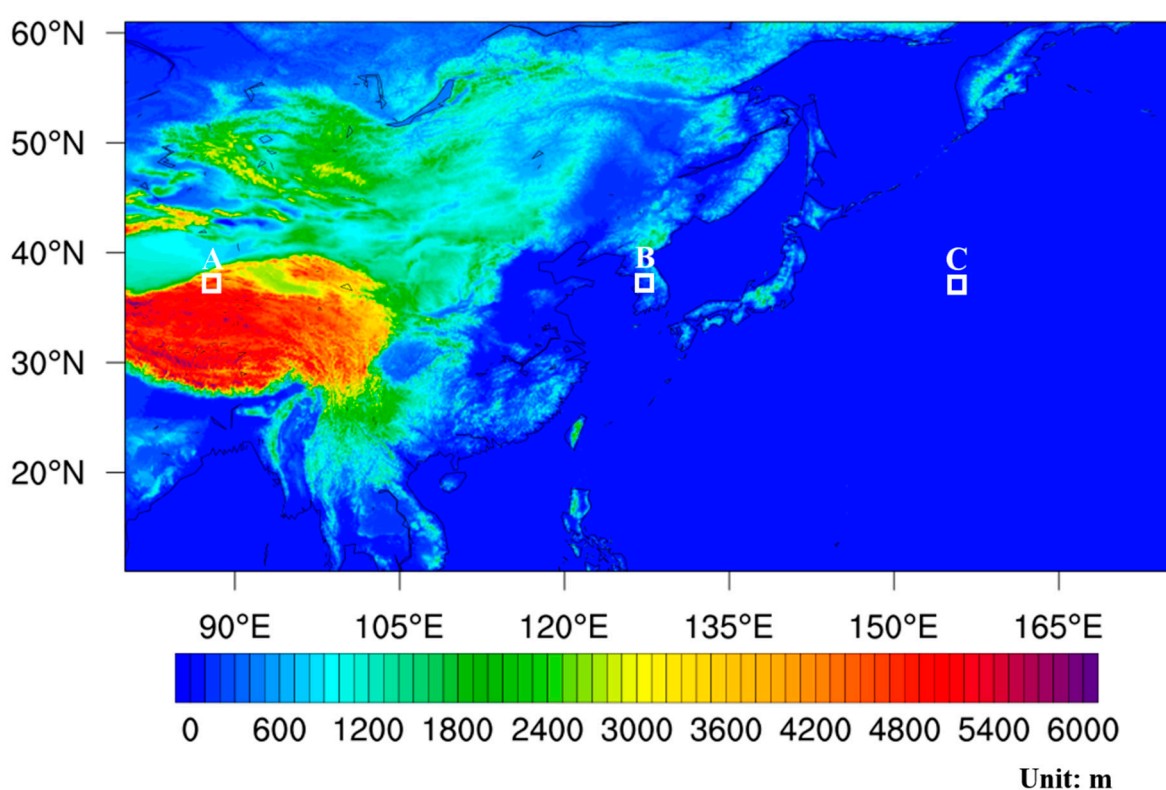

**Figure 6.** Three regions of East Asia for comparison of energy budget components. A, B, and C represent mountain, plain, and ocean, respectively.

The monthly means for each energy budget component (as seen in Figure 6) are shown in Figure 7. Since the solar radiation entering into the earth's atmosphere showed no variation over the three regions, the ISR was not included in Figure 7. The RSR at TOA, shown in Figure 7a, was significantly affected by clouds and surface albedo. Thus, the RSR for the mountain region (A) was mostly greater than that for the plain (B), and oceanic (C) regions, reaching a maximum value in April. In August, however, the RSR for the plain region (B) was greater than that for the mountain region (A) due to the presence of regional clouds, while that for the ocean (C) was highest in September and December (also due to cloud cover). As the DSR for the surface in Figure 7b was highly affected by altitude, the DSR for the mountain region (A) was greater than that for the plain or ocean regions, while the latter was greater than the plains region in August (due to cloud coverage over plains). The surface SSR in Figure 7c is a function of DSR and surface albedo, as explained above. Here, the SSR values for the mountain region in winter (covered with snow and ice) were typically lower, but higher from July–October, as snow and ice melted by the increased DSR. Since the surface albedo of plain region was lower than that of the snow-covered mountain

and ocean regions, the SSR value of this region was greater, except from July–October (the DSR and SSR values for the plain were low due to local cloud coverage during summer). In winter, the SSR values for the ocean region were lower than that for the mountain and plain regions because the ocean surface albedo was at a maximum level (due to the lower altitude of the sun).

The OLR values in Figure 7d emitted from the TOA to outer space were lower for the mountain region than for the plain or ocean regions due to lower surface temperatures (excluding during summer). The magnitude of OLR changed over plain and ocean regions, according to cloud cover. Alternatively, the surface DLR values shown in Figure 7e, were significantly influenced by atmospheric thickness and temperatures. Thus, the values for the mountain region were lower than those for the plain or ocean, while those for the plain were higher than those for the ocean in July (due to local cloud cover and precipitation). Like DLR, the values of ULR emitted by surface, shown in Figure 7f, were lower for the mountain than for the plain or ocean regions due to lower surface temperatures. On the other hand, the values of ULR for the plain were lower than that for the ocean region (excluding June and July, where land temperatures were higher). For SHF and LHF in Figure 7g,h, where wind and evaporation were important, the ocean region had significantly larger values than the mountain or plain during winter. However, these may decrease during summer periods, to less than that of the mountain and plain regions. Moreover, the values of LHF were mostly lower for the mountain than the plain or ocean regions due to the effects of dry weather, snow, and ice.

The net flux (NF_Atm) for the atmosphere of three regions in Figure 6 is shown in Figure 8. Since the atmosphere over plain was heated by absorbing radiant energy during summer and cooled by emitting radiant energy during winter, the NF_Atm for this region showed a maximum value (75.48 $W \cdot m^{-2}$) in June, and a minimum ($-74.68$ $W \cdot m^{-2}$) in December (where the positive value of NF_Atm means atmospheric warming and the negative value atmospheric cooling, respectively). This phenomenon was similar over the mountain region, but the mean difference for NF_Atm between the mountain and plain regions from April to June was about 68 $W \cdot m^{-2}$ due to the remaining snow and ice cover on the mountain. The differences of NF_Atm between the mountain and plains regions, however, were sharply decreased after July, and the NF_Atm for these regions had a similar value to that of the ocean in August. Over the ocean region, a maximum value of net flux of 292.79 $W \cdot m^{-2}$ was absorbed into the atmosphere during winter, and the net flux of $-19.29$ $W \cdot m^{-2}$ was emitted from the atmosphere in July. This large difference of NF_Atm between winter and summer in ocean region was due to SHF and LHF, as described in Figure 7g,h.

Ultimately, the annual means of net fluxes calculated for the three regions in Figure 6 are shown in Table 5. In the mountain region, the atmospheric net flux (NF_Atm) was $-15.32 \pm 3.51$ $W \cdot m^{-2}$, indicative of cooling, whereas the atmosphere over the plain and ocean absorbed the net flux of $3.11 \pm 0.12$ $W \cdot m^{-2}$ and $108.38 \pm 7.47$ $W \cdot m^{-2}$, respectively, indicative of heating (where the values after "$\pm$" refer to the standard deviations). Net flux revealed a value of $3.11 \pm 0.12$ $W \cdot m^{-2}$ of absorbed energy (atmospheric warming). As a result, a large amount of energy that was absorbed into the atmosphere over the ocean region was transported to the poles through atmospheric circulation, contributing to the energy equilibrium of earth's atmosphere.

**Table 5.** The annual mean values of net fluxes for the three regions in Figure 6, during the year of 2021. Unit of net flux: $W \cdot m^{-2}$, positive and negative sign indicates the atmospheric warming and cooling, respectively, and values after "$\pm$" refer to the standard deviations.

| Types of Region | NF_TOA | NF_Sfc | NF_Atm |
|---|---|---|---|
| Mountain | $-5.05 \pm 0.18$ | $10.27 \pm 3.33$ | $-15.32 \pm 3.51$ |
| Plain | $-1.80 \pm 0.02$ | $-4.91 \pm 0.10$ | $3.11 \pm 0.12$ |
| Ocean | $-9.88 \pm 0.09$ | $-118.26 \pm 7.38$ | $108.38 \pm 7.47$ |

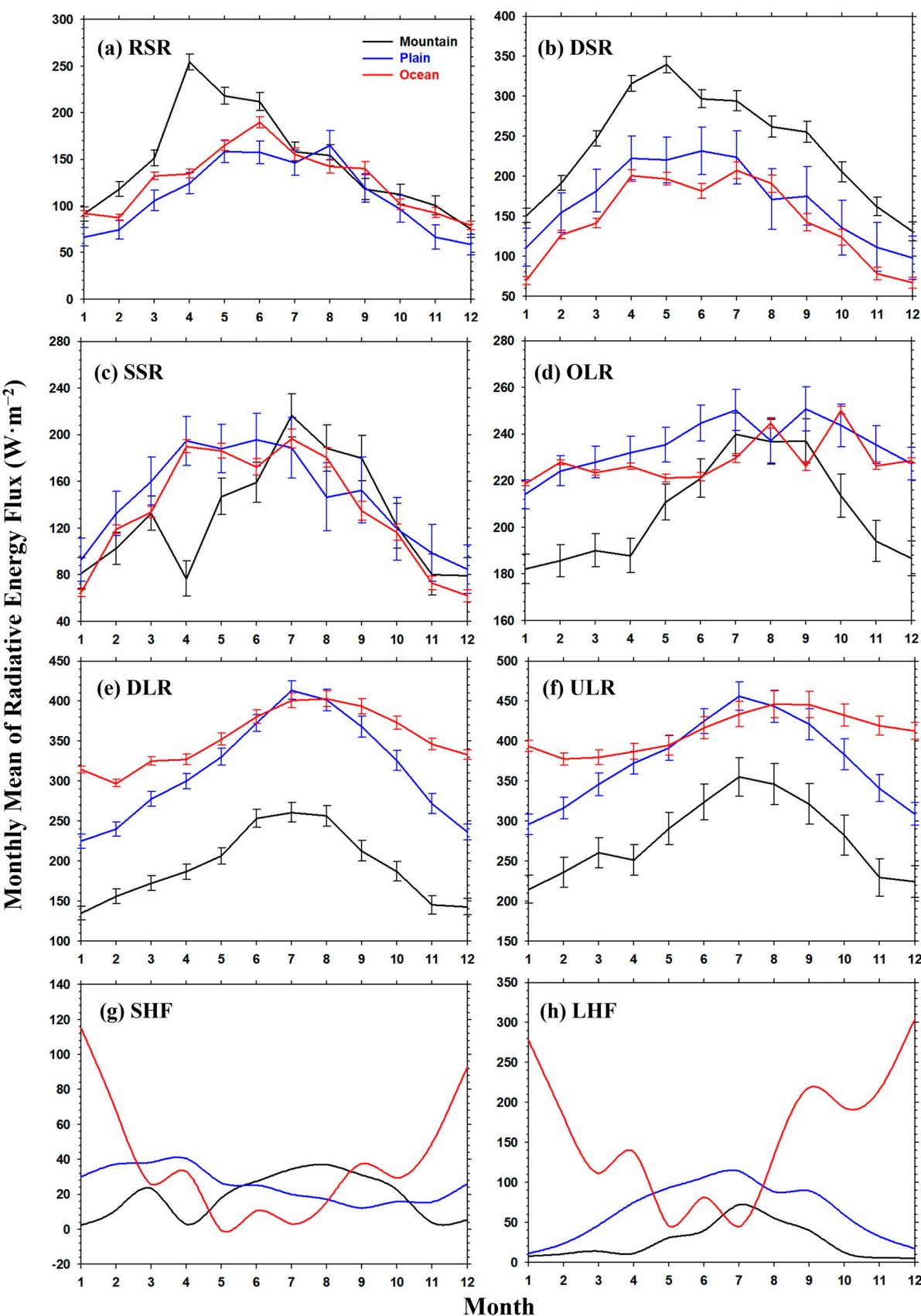

**Figure 7.** Monthly mean radiative energy budget components for the three regions shown in Figure 6 (the vertical bars in this Figure, represented the standard deviation, but their values were too small. So, they were drawn by magnifying them ten times).

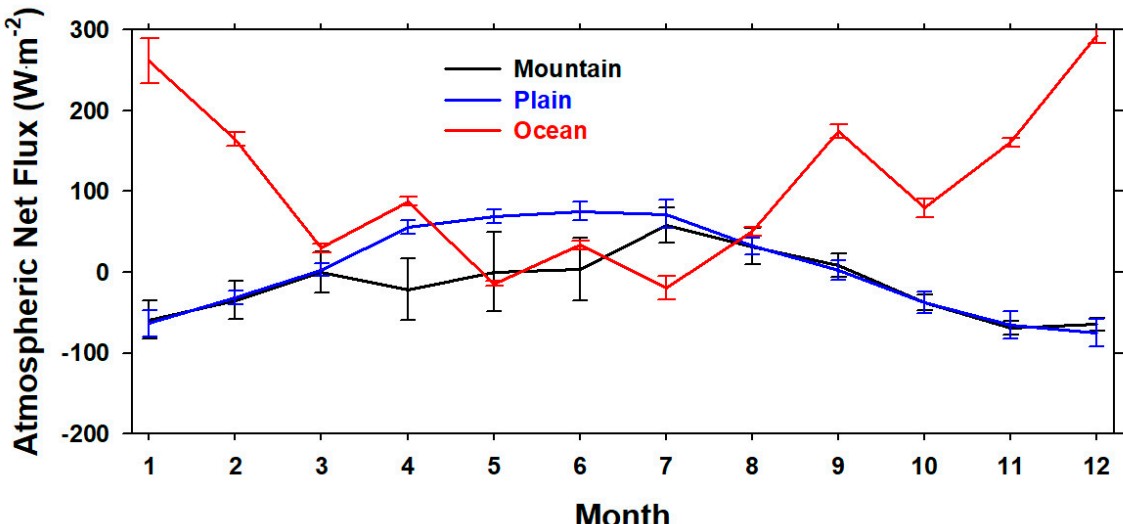

**Figure 8.** Variations in monthly mean atmosphere net flux (NF_Atm) in 2021 over mountain, plain, and ocean regions (the vertical bars in this figure, represented the standard deviation, but their values were too small. So, they were drawn by magnifying them ten times).

## 4. Conclusions

The radiative energy budget for the East Asia region was analyzed using the products of GK-2A/AMI over the year of 2021, and the conclusions are as follows:

1. **Verification of Data:** For verification of the GK-2A/AMI radiative products used in this study, they were compared with CERES single scanner footprint (SSF) Level 2 Edition 3A (Ed4A), CERES EBAF, HIMAWARI/AHI, and ERA5 reanalysis data. As a result, the statistical values between GK-2A/AMI and compared data were found to be excellent.

2. **Variation of Radiative Energy Budget components for East Asia**: During the period of this study, among the radiative energy budget components of GK-2A/AMI for East Asia, the monthly mean values of solar radiation components (ISR, RSR, DSR, and SSR) varied with solar zenith angle, and the annual mean albedo at the top of atmosphere was 0.32. However, the variations of monthly mean values for longwave radiation components (OLR, DLR, and ULR) was not distinct, but the net radiative fluxes (NF_TOA, NF_Sfc, and NF_Atm) calculated with non-radiative components (SHF and LHF) for East Asia showed the characteristics of monthly variations, and the annual mean values of their net fluxes were well matched with ERA5 data.

3. **Regional Characteristics of the Radiative Energy Budget for East Asia:** East Asia was characterized by a variety of landcovers and altitude difference. Therefore, in this study, the values of radiative energy budgets were compared for the three regions of mountain, plain, and ocean, with same latitude in East Asia. As a result, the atmospheric net flux (NF_Atm) over mountain region was $-15.32 \pm 3.51$ W·m$^{-2}$, and it was indicative of a cooling effect in the atmosphere. However, the NF_Atm over plain and ocean regions was $3.11 \pm 0.12$ W·m$^{-2}$ and $108.38 \pm 7.47$ W·m$^{-2}$, respectively, which played a role in heating up the atmosphere. Therefore, because of the large difference in atmospheric radiative energy accumulated, according to the complex topography of East Asia, the weather pattern changes for this region were significant, and the GK-2A/AMI products could be used as important data for radiative and climatic analysis in this region.

4. **Application of GK-2A/AMI radiative products:** Because the data period (of the GK-2A/AMI) used for this study was limited (one year), the results were insufficient for an accurate radiative energy budget analysis for East Asia. However, since the accuracy of the GK-2A/AMI data used in this study was found comparable to other similar types (Japanese HIMAWARI, and ERA5 reanalysis data), if this is accumulated

over several years, it could be very useful for more detailed weather and climate research of East Asia.

**Author Contributions:** I.-S.Z. wrote the manuscript and contributed to the data analysis, as well as research design; J.-B.J., K.-T.L., K.-H.L., M.-Y.L. and Y.-S.K. supervised the study, contributed to the research design, manuscript writing, and served as the corresponding authors. All authors have read and agreed to the published version of the manuscript.

**Funding:** This research was funded by National Research Foundation of Korea grant from the Korean Government (Ministry of Science and ICT—MSIT) grant number [NRF-2021M1A5A1075532].

**Data Availability Statement:** GK-2A/AMI data used for this study are available at http://datasvc.nmsc.kma.go.kr/datasvc/html/main/main.do?lang=en (accessed on 1 February 2023). The SHF and LHF data used for this study are available at https://cds.climate.copernicus.eu/cdsapp#!/dataset/reanalysis-era5-single-levels?tab=form (accessed on 1 February 2023).

**Acknowledgments:** This research was supported by a National Research Foundation of Korea grant from the Korean Government (Ministry of Science and ICT—MSIT) [NRF-2021M1A5A1075532].

**Conflicts of Interest:** The authors declare no conflict of interest.

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
