# Peer review of "Radiative Energy Budget for East Asia Based on GK-2A/AMI Observation Data"

_remotesensing, doi:10.3390/rs15061558_

Round 1

Reviewer 1 Report

Major comments:

1.     This study was analyzed using the GK-2A/AMI data to estimate atmosphere radiative energy. However, there are other datasets (e.g. CERES, HIMAWARI, and ERA5) available for estimation. The reviewer suggests that the author should enhance the strengths of the GK-2A/AMI data in introduction (Line 50). What is the advantage of using GK-2A/AMI data?

2.     This study tries to verify the GK-2A/AMI radiative products based on CERES, HIMAWARI, and ERA5 data. However, the comparison's statistical methods (R, MBE, MPE, RMSE) are not consistent in each dataset. Suggested to combine Table 2, Table 3, and Table 4 into a single table and use the same statistical methods. This will help the reader to compare each dataset.

3.     The global radiative energy budget by ERA5 is 0.89, but the value “in the East Asia region” from this study is -4.09 by GK-2A/AMI. This value really depends on the research area which did not particularly clearly defined (only shown in Line 47) in the manuscript. Suggested to explain clearly the “East Asia region bound” in Abstract and Chapter 2 (Materials and Methods).

4.     The radiative energy estimate has its uncertainty. Please include the uncertainty information in the statistical result. Suggested adding the standard deviation in Table 7, and Figure 4, Figure 7, Figure 8.

Minor comments:

1.     In the abstract, please strength (point out/add) the advantage of GK-2A/AMI specifically.

2.     Line 100-101: Please remove the comma in the end of Eq.1 and Eq. 2.

3.     Line 108-109: Please explain the abbreviation of “LST” and “SST”.

4.     Line 120-122: The reason for the DSR and ASR difference between CERES and GK-2A/AMI was unclear.

5.     Line 153-155: “the value for RMSE (R)… more than 0.97” The sentence was unclear. The value of “0.97” was R but not RMSE, and the value “4 Wm-2” and “5%” didn’t show in Table 3.

Author Response

attached file

Reviewer 2 Report

In this paper, the authors analyzed the radiative energy budget for East Asia, during the year 2021 using the GK- 2A/AMI and ERA5 data. It is meaningful for understanding the radiative energy budget in special areas. Some advices and suggestions are as follows:

1.        Line 84 Try to use another abbreviation for absorbed shortwave radiation at surface, since ASR is usually referred to the radiation at TOA. This may confuse the readers.

2.        Line 114 Why not use CERES SSF Ed4A instead of Ed3A. Large refinements have been performed in the new edition. The Edition4A SW fluxes are between 1 and 2 W lower than Edition3A.

3.        Line 116-118 More data processing details are need. 20 km x 20 km are only the nominal resolution of CERES SSF data. How did you make sure the two datasets are spatially collocated.

4.        Table 3 and Table 4 Why not show RMSE values.

5.        Line 179 manuscript/paper

6.        Equation 13 The definition of surface net flux is not right. SHF and LHF are two components into which the net flux is partitioned. I suggest to remove the SHF and LHF.

7.        Table 5 No need to show the results of every month. Seasonal average results would be enough and clearer to the readers. CERES data should also be presented in this table for comparisons.

8.        Line 211 “where the negative value indicated an upward direction and so does the rest of this manuscript” should be mentioned earlier in Line 184.

9.        Line 297 The authors mention “The annual mean values for the regional characteristics analysis of each radiative products shown in Table 5, were presented in Figure 5.” Then why not move this part after the description of Table 5.

Author Response

attached file

Round 2

Reviewer 2 Report

All my concerns are well addressed and I do not have further comments.